# Evaluation of an optimized germline exomes pipeline using BWA-MEM2 and Dragen-GATK tools

**Nofe Alganmi**[1,2]◎*, **Heba Abusamra**[2]◎

**1** Department of Computer Science, Faculty of Computing and Information Technology, King Abdulaziz University, Jeddah, Saudi Arabia, **2** Center of Excellence in Genomic Medicine Research (CEGMR), King Abdulaziz University, Jeddah, Saudi Arabia

◎ These authors contributed equally to this work.
* nalghanimi@kau.edu.sa

**Data Availability Statement:** The data has successfully uploaded to an online repository. DOI: 10.5061/dryad.6m905qg2s URL: https://datadryad.org/stash/share/dtGgC6i108cuGOxF1IzexBABG6F7kpa3lYQO5NeAkNc Data reference: Alganmi, nofe

## Abstract

The next-generation sequencing (NGS) technology represents a significant advance in genomics and medical diagnosis. Nevertheless, the time it takes to perform sequencing, data analysis, and variant interpretation is a bottleneck in using next-generation sequencing in precision medicine. For accurate and efficient performance in clinical diagnostic lab practice, a consistent data analysis pipeline is necessary to avoid false variant calls and achieve optimum accuracy. This study aims to compare the performance of two NGS data analysis pipeline compartments, including short-read mapping (BWA-MEM and BWA-MEM2) and variant calling (GATK-HaplotypeCaller and DRAGEN-GATK). On Whole Exome Sequencing (WES) data, computational performance was assessed using several criteria, including mapping efficiency, variant calling performance, false positive calls rate, and time. We examined four gold-standard WES data sets: Ashkenazim father (NA24149), Ashkenazim mother (NA24143), Ashkenazim son (NA24385), and Asian son (NA25631). In addition, eighteen exome samples were analyzed based on different read counts, and coverage was used precisely in the run-time assessment. By using BWA-MEM 2 and Dragen-GATK, this study achieved faster and more accurate detection for SNVs and indels than the standard GATK Best Practices workflow. This systematic comparison will enable the bioinformatics community to develop a more efficient and faster solution for analyzing NGS data.

## Introduction

The development of the next generation of high-throughput next-generation sequencing (NGS) platforms and a reliable and consistent method of identifying genetic variants have made it possible to use personal genome information to detect and identify patients' genetic variations and etiological factors of diseases. However, the analysis and interpretation of these large-scale sequencing data continue to be challenging regarding accurate variant detection, including SNVs and indels, and turnaround time efficiency.

The sequence mapping and variant calling techniques available today are diverse and can be incorporated into diverse pipelines. An NGS pipeline typically includes an aligner and a

(2023), Genati labs GIAB dataset, Dryad, Dataset, https://doi.org/10.5061/dryad.6m905qg2s.

**Funding:** The funders had no role in study design, data collection and analysis, decision to publish, or preparation of the manuscript.

**Competing interests:** The authors have declared that no competing interests exist.

variant caller: the aligner maps sequencing reads to a reference genome, and the variant caller identifies variant sites and genotypes. The performances of different aligners have been studied extensively [1–4]. A widely used tool for aligning sequencing reads to a reference genome is the Burrows-Wheeler Aligner (BWA) [5]. There are three versions of BWA: BWA-backtrack, BWA-SW, and BWA-MEM. Each has been designed to handle sequence read problems; for example, BWA-backtrack is better for short sequences, BWA-SW is more sensitive to frequent gaps in the reads, while BWA-MEM is preferred for 70bp reads and more. In BWA-MEM2, the newest version of BWA-MEM, the alignment is identical to BWA, but the runtime is 1-3 times faster depending on the use case, dataset, and the machine [6].

There have been many different variants calling algorithms developed over the years [7–9]. When calling variants, the key challenge is distinguishing between false positives and true variants (due to contamination, library preparation artifacts, sequencing artifacts, mismapping). Indel errors are especially difficult to detect because they are more likely to occur in tandem repeats. The probability of detecting the error depends on the period and the repeat length. A widely used variant calling algorithm is GATK-HaplotypeCaller (GATK-HC) from Genome Analysis ToolKit GATK [10]. This algorithm uses a Hidden Markov Model (HMM) to correctly model errors as part of the probability calculation, using predetermined functions that do not depend on individual samples.

A new collaboration that brings together the GATK team and the DRAGEN team (an Illumina company) to co-develop analysis methods and pipelines provides enhanced accuracy in the open-source software version (labeled DRAGEN-GATK). By adding sample-specific logic before variant calling and explicitly considering STR period and length within a new technique called DragSTR, DRAGEN could model the indel error process more accurately. This step, known as sample-specific logic, is part of the auto-calibration step used to find the probability of indel errors and variants based on the BAM input. Their testing confirmed better detection performance in terms of sensitivity and precision: Previously missed calls were recovered, and false positives were removed.

It is hence crucial to evaluate variant calling pipelines that combine aligners and variant callers with the optimal combination that can yield accurate variant calls including SNVs and indels. Our aim in this study was to evaluate the performance of the optimized BWA-MEM2 alignment tool and the optimized Dragen-GATK variant caller tool for detecting SNVs and indels separately. In order to accomplish this objective, we used human whole exome sequencing data from Genome in a Bottle (GiaB) high-confidence GRCh37 reference sets from Ashkenazim father (NA24149), Ashkenazim mother (NA24143), Ashkenazim son (NA24385), and Asian son (NA24631). The time performance comparison was also conducted using eighteen exome samples collected from different read counts and coverage levels since the four reference samples from GiaB were approximate of the same size in yields and read counts perspective. All the analyses were performed on the supercomputer Aziz. Several performance statistics are reported here with respect to execution time, recall, and precision.

## Materials and methods

### Datasets

As a gold standard practice to validate the whole exome sequencing pipelines performance, The highly confident variant call-sets in the Variant Call Format (VCF) of the Chinese son (HG005_NA24631), and Ashkenazi Jewish trio sets consist of the mother (HG004_NA24143), father (HG003_NA24149) and son (HG002_NA24385) using hg19 coordinates, were downloaded from the Genome in a Bottle (GIAB) website (https://ftp-trace.ncbi.nlm.nih.gov/giab/ftp/release/).

In addition, we collected real data from eighteen patients who were sequenced at Genati labs (which is a College of American Pathologists accredited laboratory) in the Center of Excellence in Genomic Medicine Research (CEGMR) at King Abdulaziz University. This study was approved by IRB no. 32-CEGMR-Bioeth-2021. Since the four reference samples from GiaB were of the same size in yields and read counts perspective, we selected the eighteen exome sequencing files to be of different yields, coverage and mapped reads to evaluate the pipelines' time performance. To facilitate the time performance comparison, these samples were distributed into four groups based on fastq data yield. Six samples were of the smallest amount yield (2-3 GB), eight samples of 4-6 GB yield, 3 samples of 8-11 GB yield, and one sample with the largest yield of 17 GB. Table 1, provides detailed information on these eighteen data sets grouped based on fastq data yield, and passing filter (PF) clusters for both lane 1 and lane 2.

## Exome sequencing

All samples in the study were sequenced using Illumina Novaseq 6000 sequencing platform to produce 2x101 paired-end reads. The Illumina DNA Prep with Enrichment library was used to capture the sequences of 214,126 targeted exonic regions. The start and stop of chromosome locations in GRCH37/hg19 were downloaded from the Illumina website: (https://support. illumina.com/downloads/nextera-dna-exome-product-files.html). The GIAB and the 18 fastq files contained more than 94% bases with quality that is higher than Q30.

## WES bioinformatics pipelines

The first pipeline used in the comparison was the GATK Best Practices workflow (https:// software.broadinstitute.org/gatk/best-practices) for analyzing NGS data [11]. We started by trimming the raw paired-end reads represented in FASTQ files by removing the low-quality reads and adapter sequences with trimmomatic (version 0.36). Then, FastqQC (version 3) was applied to evaluate the quality of the fastq data obtained. The raw paired-end reads were

**Table 1. The eighteen exome sequencing data sets description.**

| Sample ID | FASTQ data yield | Number of sample | Lane 1 PF Cluster | Lane 2 PF Cluster |
|---|---|---|---|---|
| Sample 1 | 2-3 GB | 6 samples | 13,697,078 | 13,417,631 |
| Sample 2 | | | 13,977,538 | 13,736,079 |
| Sample 3 | | | 14,417,563 | 14,227,541 |
| Sample 4 | | | 14,717,874 | 14,536,563 |
| Sample 5 | | | 15,285,614 | 15,094,139 |
| Sample 6 | | | 16,620,551 | 16,436,174 |
| Sample 7 | 4-7 GB | 8 samples | 19,826,361 | 19,403,315 |
| Sample 8 | | | 19,892,847 | 19,605,455 |
| Sample 9 | | | 20,818,858 | 20,572,995 |
| Sample 10 | | | 21,377,034 | 21,204,559 |
| Sample 11 | | | 26,074,597 | 25,724,437 |
| Sample 12 | | | 38,285,445 | 37,909,674 |
| Sample 13 | | | 39,393,585 | 39,124,595 |
| Sample 14 | | | 39,546,390 | 38,882,944 |
| Sample 15 | 8-11 GB | 3 samples | 43,615,010 | 43,271,221 |
| Sample 16 | | | 45,173,737 | 44,612,342 |
| Sample 17 | | | 50,489,897 | 49,841,076 |
| Sample 18 | 17 GB | 1 samples | 87,401,725 | 86,818,370 |

aligned with GRCH37 reference genome using BWA-mem aligner (version 0.7.12), to generate BAM files, which were then sorted by genome position using samtools (version 1.2), then marked for duplicates using picard-tools (version 2.2.1). The raw BAM files were refined by BQSR using default parameters for GATK (version 4.1.2). The next step is to perform variant calling (SNVs and indels) using the HaplotypeCaller module and filter the VCFs based on (allele frequency, read depth, mapping quality, and variant quality score). A report with mapping efficiency, variant calling performance, false positive call rate, and time is generated at the end.

An optimized pipeline has been proposed as the second pipeline. This pipeline follows the same workflow structure that is described in the GATK workflow best practices. However, a recently upgraded and improved aligner (BWA-MEM2) and variant caller (Dragen-GATK) were utilized. The optimized pipeline starts with a trimming adapter, and low-quality reads using trimmomatic (version 0.36) and FASTQC (Version 3) are used to assess raw data quality. Using BWA-MEM2 (version 2.2.1), the reads were aligned to GRCH37. Post-alignment processing was performed with samtools (version 1.2) to transform sam files into bams, sorting and creating indexes on bams, and with GATK (version 4.2.2) for marking duplicates, base recalibration, and variant calling. The pipeline ended with vcf filtration. We also utilized the spark-enabled tools available in GATK version 4 to optimize performance and CPU utilization [12]. The two pipelines are used in the comparison study as described in Fig 1.

## Computational resources

Both pipelines were run using a job array on King Abdulaziz University's High-Performance Computing Center (http://hpc.kau.edu.sa). AZIZ is equipped with Fujitsu PRIMERGY CX400, Intel True Scale QDR, and Intel Xeon E5-2695v2 12C 2.4GHz processors with 256 GB RAM. Each pipeline had the same number of nodes, CPUs, and samples in the job arrays.

## Pipeline comparison metrics

To evaluate the variant callers' performance, Illumina Variant Calling Assessment Tool v4.0.4 (VCAT) was used to compare variants generated by the pipelines with standard variants provided by the GiaB consortium. Only variants covered in the exome panel v1.2 were included in the comparison. For each type of variant (SNVs and indels) the following values have been calculated (for all datasets): number of variants in GiaB and detected by the pipeline as true positives (TP), number of variants detected by the pipeline but not found in GiaB as false positives (FP), and number of variants in GiaB that are not detected by the pipeline as false negatives (FN). Finally, the variant calling pipeline was evaluated using the following metrics: sensitivity = TP / (TP + FN), precision = TP / (TP + FP), and F-score = 2TP / (2 TP + FP + FN).

After that, the mapping efficiency was compared using the quality control reports generated for each sample from both of the pipelines. The metrics used for the comparison are: total targeted reads, total targeted bases, and mean target coverage.

In a final step, the run time of both pipelines was measured. To better assess the run time efficiency, the run time evaluation procedure was divided into three phases: (1) an overall run time performance from the beginning to the end of the pipeline,(2) upstream analysis (Fastq to BAM file), and (3) downstream analysis (BAM to VCF file).

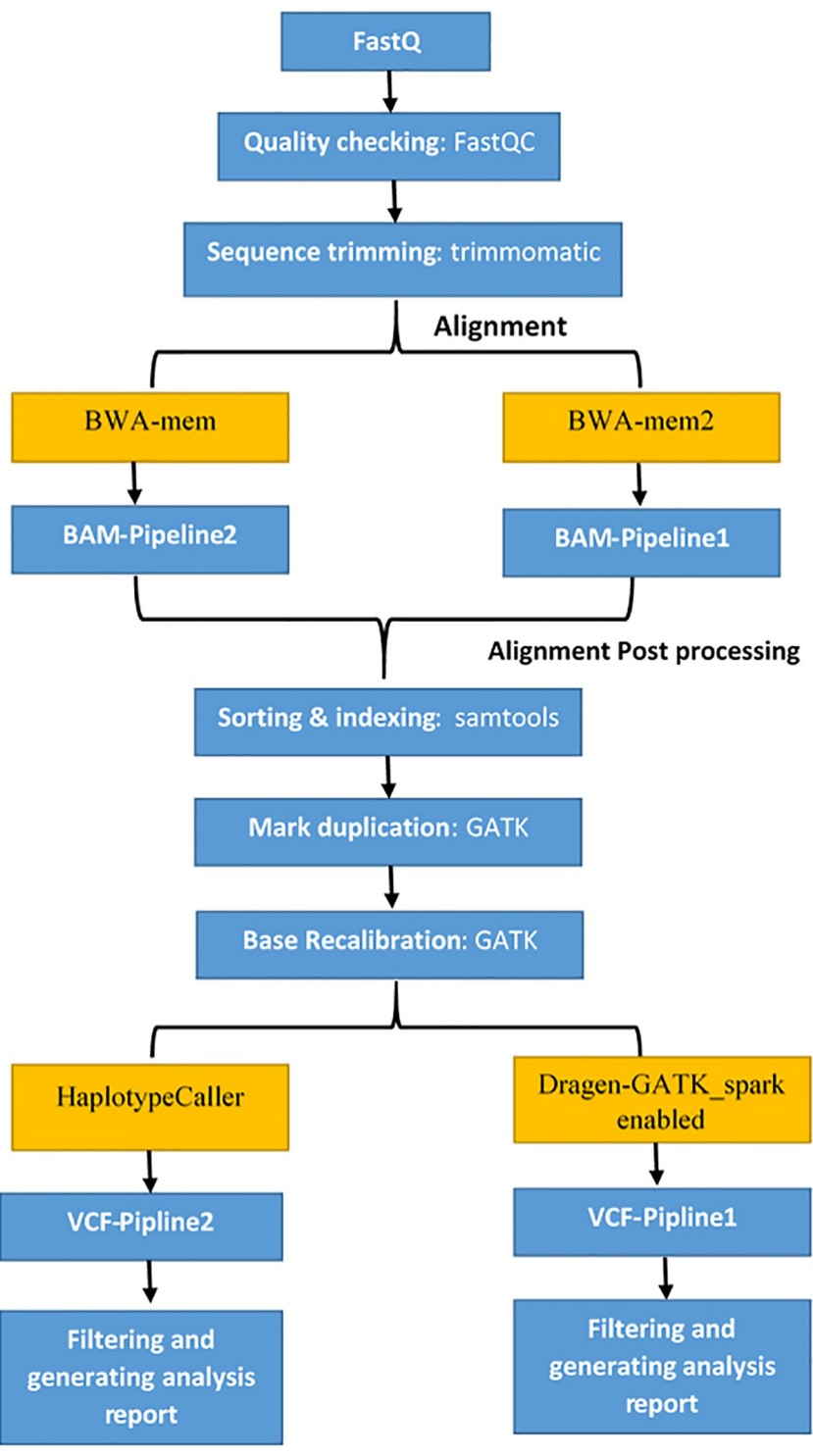

**Fig 1. Bioinformatics pipelines.**

## Results

### Variant calling and false positive calls performance assessment

The pipelines were evaluated by calculating true positive (TP), false positive (FP), false negative (FN) variants, precision, and recall using VCAT v4.0.4. Results are presented in Table 2 for SNVs and indels. As for SNVs variant calls, both pipelines demonstrated good performance, with accuracy and recall exceeding 97% and 98%, respectively. The BWA-MEM 2+Dragen-GATK showed higher performance in the sample NA24149 where recall grew from 97% to 99% as a result of true positives and false negatives increasing by 9% and decreasing by 5%, while its performance was slightly improved in the other samples.

Due to the challenging nature of indel calls, BWA-MEM 2+DRAGEN-GATK achieves better performance than GATK best practice workflow in terms of term recall, which was increased by a minimal 1% since the difference in the number of true positives was small between the callers. The number of false positives varied greatly between BWA-MEM 2+DRAGEN-GATK and GATK best practice workflow (Fig 2), which contributed to an increase in indels precision to 95%.

As the BWA-MEM 2+Dragen-GATK pipeline generated reliable results, there is a need to ensure that the results are not just reliable but even reproducible with every run analysis. For that, we assessed the reproducibility of the variants (SNVs/indels) called by the BWA-MEM 2 +Dragen-GATK pipeline using the GIaB samples in four replicated runs. The performance results of the four replicated results for each sample were identical Table 3. Furthermore, we assessed deeply the variant detection of the four replicates. We observed high concordance of the variants among replicated runs (99.9%) while the error rate was (0.019%),(0.002%), (0.09%), (0.07%) in the NA24149, NA24143, NA24385,and NA24631 samples (Fig 3).

### Run time performance assessment

This study evaluates the run-time performance of BWA-MEM + Dragen-GATK against the GATK Best Practice pipeline. A total of 18 short-read datasets sampled from different coverage and read counts were analyzed to determine the impact on computation time. In both pipelines, the computation time correlated linearly with the sample read count. Each sample run

**Table 2. Performance comparison: Numbers of true positive and false positive, precision, and recall of SNVs/indels for the two pipelines.**

| Pipeline | Sample | SNV TP | SNV FP | SNV FN | Indel TP | Indel FP | Indel FN | SNV Precision | SNV Recall | Indel Precision | Indel Recall |
|---|---|---|---|---|---|---|---|---|---|---|---|
| **BWA-MEM +HaplotypeCaller** | **NA24149** | 25,685 | 383 | 586 | 1,399 | 574 | 304 | 98.53% | 97.77% | 70.95% | 82.15% |
| **BWA-MEM 2+Dragen-GATK** | | 26,032 | 340 | 239 | 1,464 | 93 | 239 | 98.71% | 99.09% | 94.03% | 85.97% |
| **BWA-MEM +HaplotypeCaller** | **NA24143** | 26,166 | 329 | 205 | 1,569 | 406 | 212 | 98.76% | 99.22% | 79.47% | 88.10% |
| **BWA-MEM 2+Dragen-GATK** | | 26,170 | 327 | 201 | 1,591 | 82 | 190 | 98.77% | 99.24% | 95.01% | 89.33% |
| **BWA-MEM +HaplotypeCaller** | **NA24385** | 26,318 | 368 | 179 | 1,574 | 475 | 199 | 98.62% | 99.32% | 76.85% | 88.78% |
| **BWA-MEM 2+Dragen-GATK** | | 26,322 | 348 | 175 | 1,587 | 78 | 186 | 98.69% | 99.34% | 95.32% | 89.51% |
| **BWA-MEM +HaplotypeCaller** | **NA24631** | 26,743 | 321 | 194 | 1,447 | 241 | 144 | 98.81% | 99.28% | 85.71% | 90.95% |
| **BWA-MEM 2+Dragen-GATK** | | 26,745 | 325 | 192 | 1,450 | 70 | 141 | 98.80% | 99.29% | 95.39% | 91.14% |

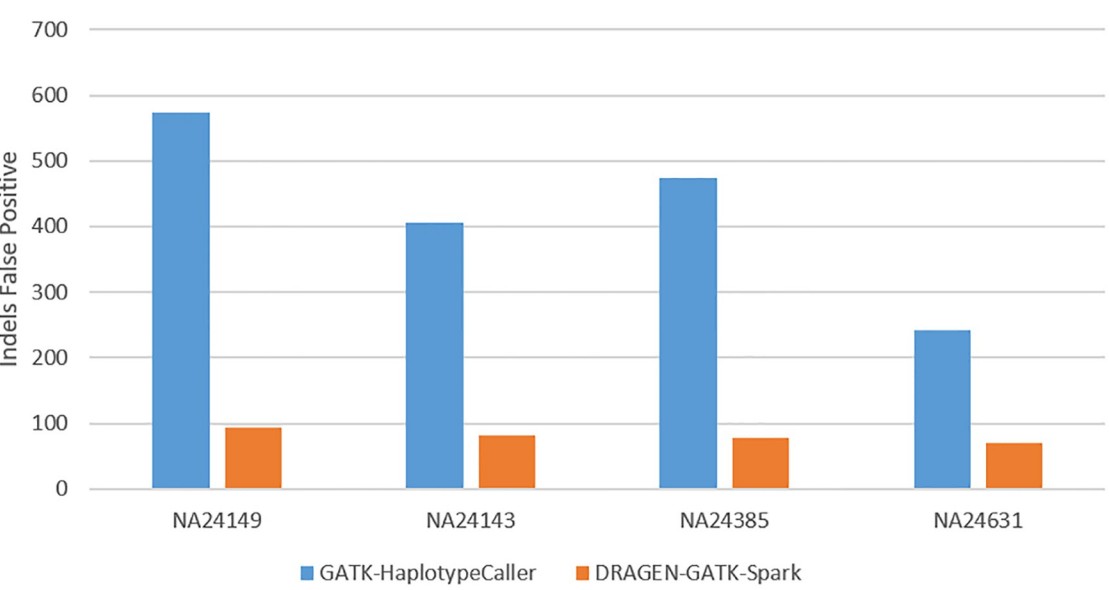

**Fig 2. Performance comparison: Indels false positive calls.**

under each pipeline 20 times, and the average of the time performance was calculated as shown in Table 4. Overall, the BWA-MEM2+DRAGEN-GATK pipeline was much more efficient compared to the standard GATK pipeline, producing a half-time reduction in total analysis time for all samples of different yields (Fig 4).

To further assess the run time differences between the two pipelines, we compared the run time at different phases of pipeline 1. Aligners run time (i.e.time required to generate sorted, marked duplicate Bam file from the fastq file), and 2. variant callers run time (the time required to generate the original and filtered VCF file from the BAM file). Fig 5A shows the

**Table 3. BWA-MEM2+Dragen-GATK reproducibility performance comparison: Numbers of true positive and false positive, precision, and recall of SNVs/indels.**

| Sample | Run# | SNV TP | SNV FP | SNV FN | Indel TP | Indel FP | Indel FN | SNV Precision | SNV Recall | Indel Precision | Indel Recall |
|---|---|---|---|---|---|---|---|---|---|---|---|
| NA24149 | Run1 | 26,032 | 340 | 239 | 1,464 | 93 | 239 | 98.71% | 99.09% | 94.03% | 85.97% |
|  | Run2 | 26,032 | 340 | 239 | 1,464 | 93 | 239 | 98.71% | 99.09% | 94.03% | 85.97% |
|  | Run3 | 26,032 | 340 | 239 | 1,464 | 93 | 239 | 98.71% | 99.09% | 94.03% | 85.97% |
|  | Run4 | 26,032 | 340 | 239 | 1,464 | 93 | 239 | 98.71% | 99.09% | 94.03% | 85.97% |
| NA24143 | Run1 | 26,170 | 327 | 201 | 1,591 | 82 | 190 | 98.77% | 99.24% | 95.01% | 89.33% |
|  | Run2 | 26,170 | 327 | 201 | 1,591 | 82 | 190 | 98.77% | 99.24% | 95.01% | 89.33% |
|  | Run3 | 26,170 | 327 | 201 | 1,591 | 82 | 190 | 98.77% | 99.24% | 95.01% | 89.33% |
|  | Run4 | 26,170 | 327 | 201 | 1,591 | 82 | 190 | 98.77% | 99.24% | 95.01% | 89.33% |
| NA24385 | Run1 | 26,322 | 348 | 175 | 1,587 | 78 | 186 | 98.69% | 99.34% | 95.32% | 89.51% |
|  | Run2 | 26,322 | 348 | 175 | 1,587 | 78 | 186 | 98.69% | 99.34% | 95.32% | 89.51% |
|  | Run3 | 26,322 | 348 | 175 | 1,587 | 78 | 186 | 98.69% | 99.34% | 95.32% | 89.51% |
|  | Run4 | 26,322 | 348 | 175 | 1,587 | 78 | 186 | 98.69% | 99.34% | 95.32% | 89.51% |
| NA24631 | Run1 | 26,745 | 325 | 192 | 1,450 | 70 | 141 | 98.80% | 99.29% | 95.39% | 91.14% |
|  | Run2 | 26,745 | 325 | 192 | 1,450 | 70 | 141 | 98.80% | 99.29% | 95.39% | 91.14% |
|  | Run3 | 26,745 | 325 | 192 | 1,450 | 70 | 141 | 98.80% | 99.29% | 95.39% | 91.14% |
|  | Run4 | 26,745 | 325 | 192 | 1,450 | 70 | 141 | 98.80% | 99.29% | 95.39% | 91.14% |

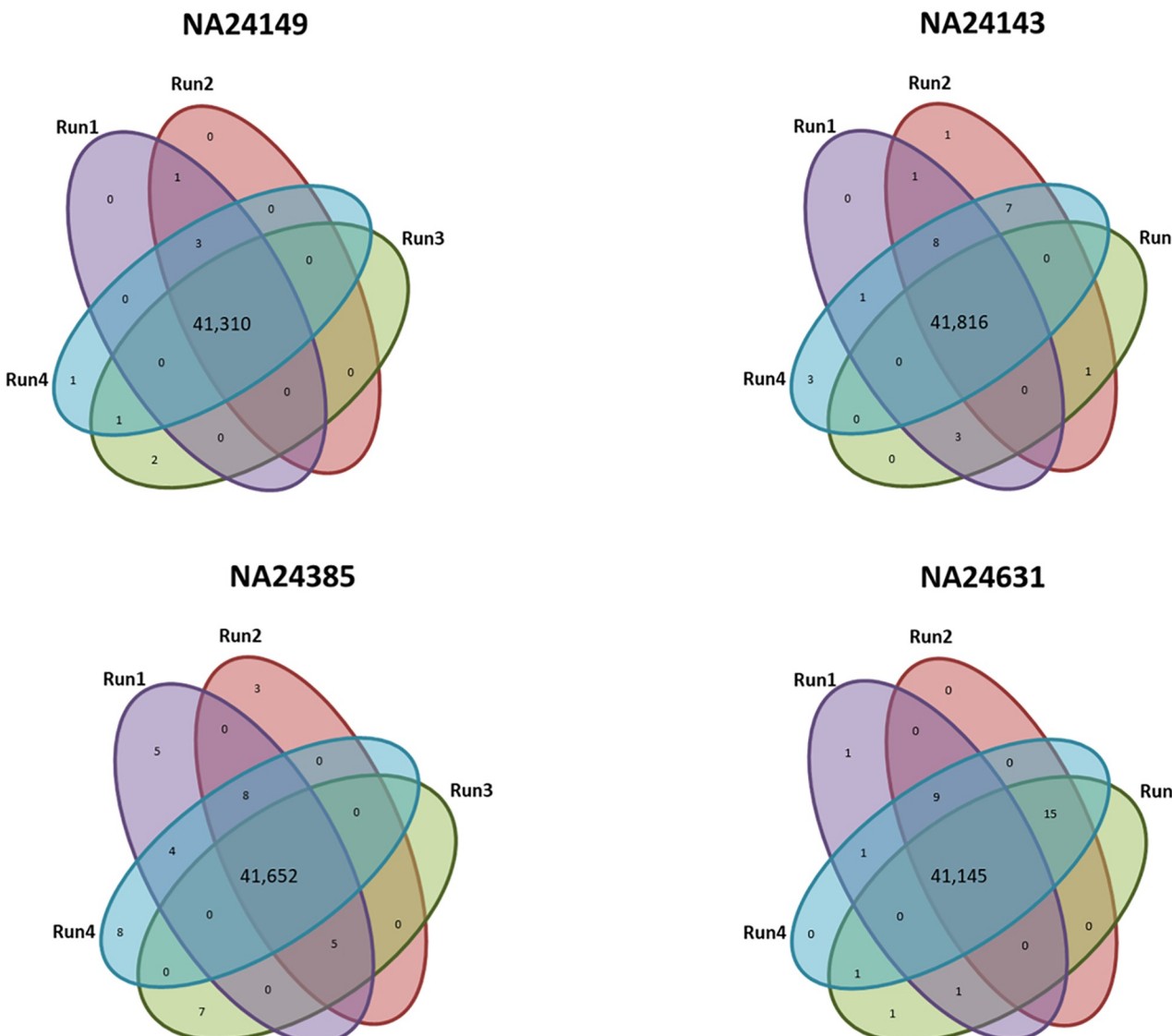

**Fig 3. Venn diagrams showing the intersection of variants called among replicate runs using BWA-MEM2+Dragen-GATK pipeline on NA24149, NA24143, NA24385,and NA24631 samples.**

time performance of BWA-MEM2 with the advantage of being more than 2X faster than its precedence, BWA-MEM. To further evaluate mapping results from the two tools and if any discrepancies could impact downstream genetics variants detection, three metrics (total targeted reads, total targeted bases, and mean target coverage) were used to compare the mapping results from the two tools and equivalent mapping results as were achieved (Fig 5B).

Fig 6 shows the time performance of the two variant callers. Dragen-GATK could reduce the variant calling time to half compared to the GATK-haplotypecaller.

## Discussion

One significant challenge when performing whole exome sequencing (WES) is how to process the data to detect the exact genetic variants that cause the disease. Alignment and variant

**Table 4. The total run time performance per hours of the two pipelines for each sample.**

|  | FASTQ data yield (MB) | BWA-MEM+HaplotypeCaller | BWA-MEM 2+ Dragen-GATK-SPRK |
|---|---|---|---|
| **Sample 1** | 2,767 | 7:44 | 2:44 |
| **Sample 2** | 2,823 | 7:40 | 2:46 |
| **Sample 3** | 2,912 | 7:57 | 3:04 |
| **Sample 4** | 2,973 | 8:06 | 2:53 |
| **Sample 5** | 3,088 | 8:00 | 2:59 |
| **Sample 6** | 3,357 | 8:48 | 3:12 |
| **Sample 7** | 4,005 | 9:28 | 3:50 |
| **Sample 8** | 4,018 | 9:09 | 3:31 |
| **Sample 9** | 4,205 | 9:30 | 3:51 |
| **Sample 10** | 4,318 | 9:39 | 3:42 |
| **Sample 11** | 5,267 | 10:35 | 4:36 |
| **Sample 12** | 7,734 | 13:43 | 6:07 |
| **Sample 13** | 7,958 | 13:42 | 6:12 |
| **Sample 14** | 7,988 | 13:40 | 6:05 |
| **Sample 15** | 8,810 | 14:27 | 6:50 |
| **Sample 16** | 9,125 | 14:53 | 6:40 |
| **Sample 17** | 10,199 | 16:34 | 8:09 |
| **Sample 18** | 17,655 | 25:39 | 12:54 |

calling tools are essential to this process. In this study, we evaluated the performance of three short-read sequence alignment algorithms (BWA-MEM, BWA-MEM2) and variant calling algorithms (GATK-HC, DRAGEN-GATK) on whole exome sequencing data. Evaluation criteria included mapping efficiency, variant calling performance, false positive calls, and time

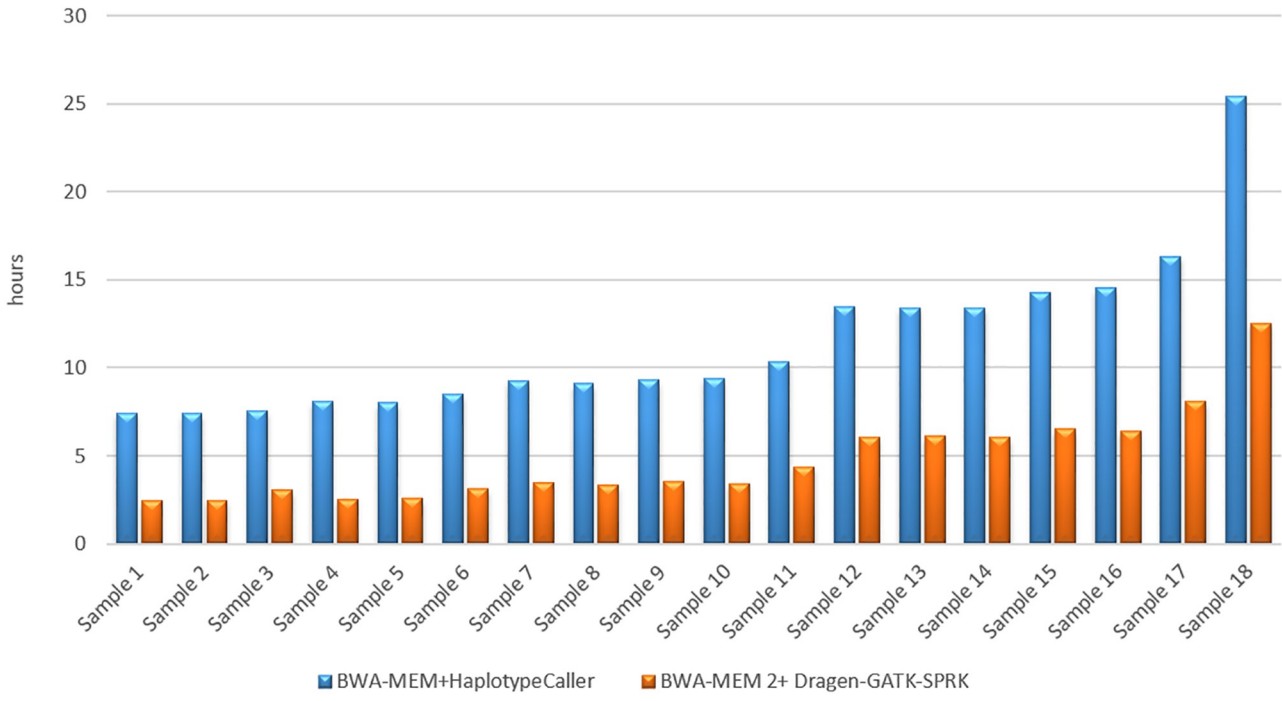

**Fig 4. Run time performance of the two aligners.**

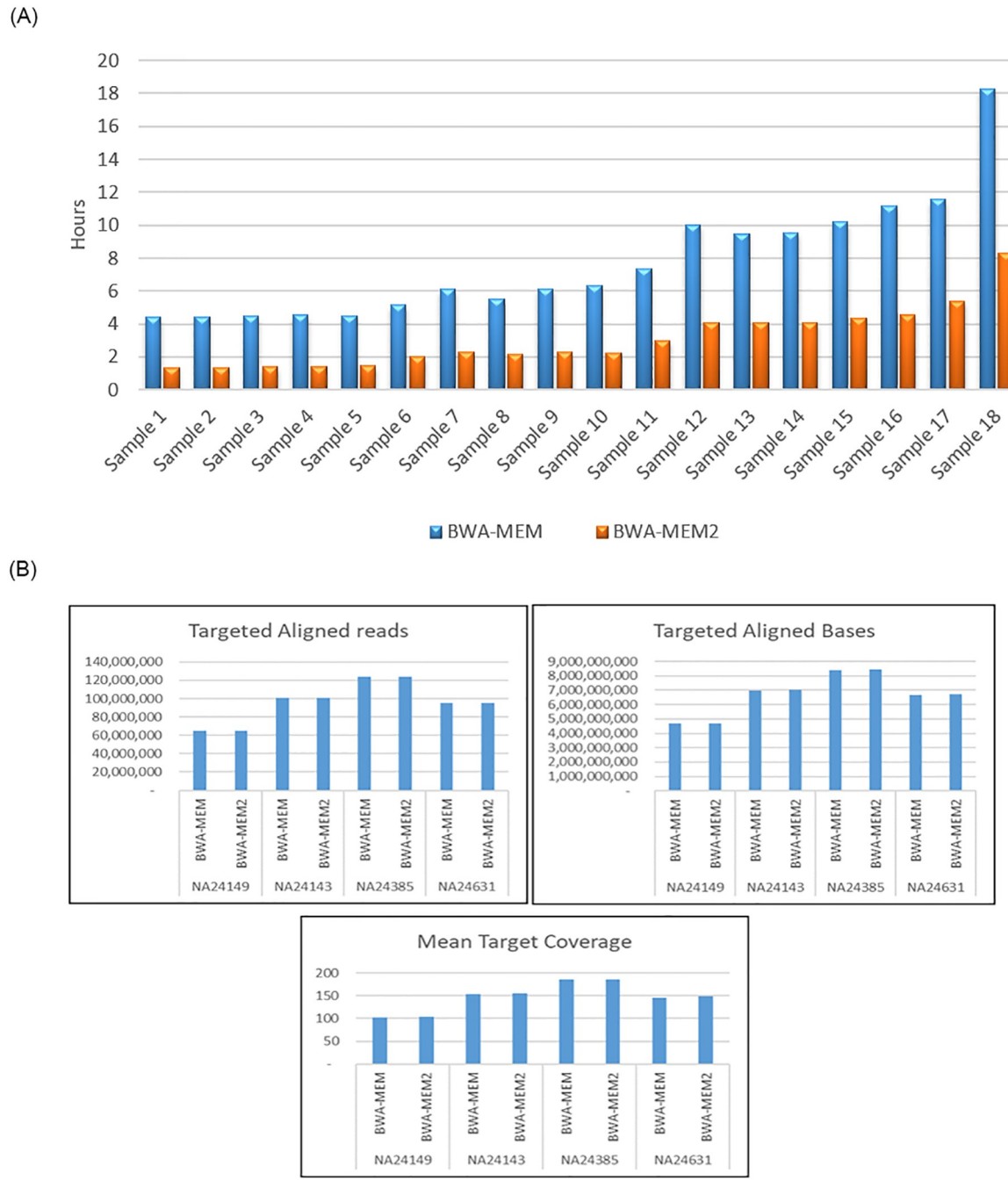

**Fig 5A. A.** Run time performance of the two aligners. **B**. Mapping efficiency metrics (total targeted reads, total targeted bases, and mean target coverage) between BWA-mem and BWA-mem2 aligners.

performance. The performance assessment used a gold-standard set of exomes taken from four Ashkenazim fathers (NA24149), Ashkenazim mothers (NA24143), Ashkenazim sons (NA24385), and Asian sons (NA24631), along with real-world data collected from different read counts and coverage. Samples were sequenced on a Novaseq 6000 sequencing platform (Illumina) with 2 * 101 cycles reads mapped to the hg19 reference human genome assembly

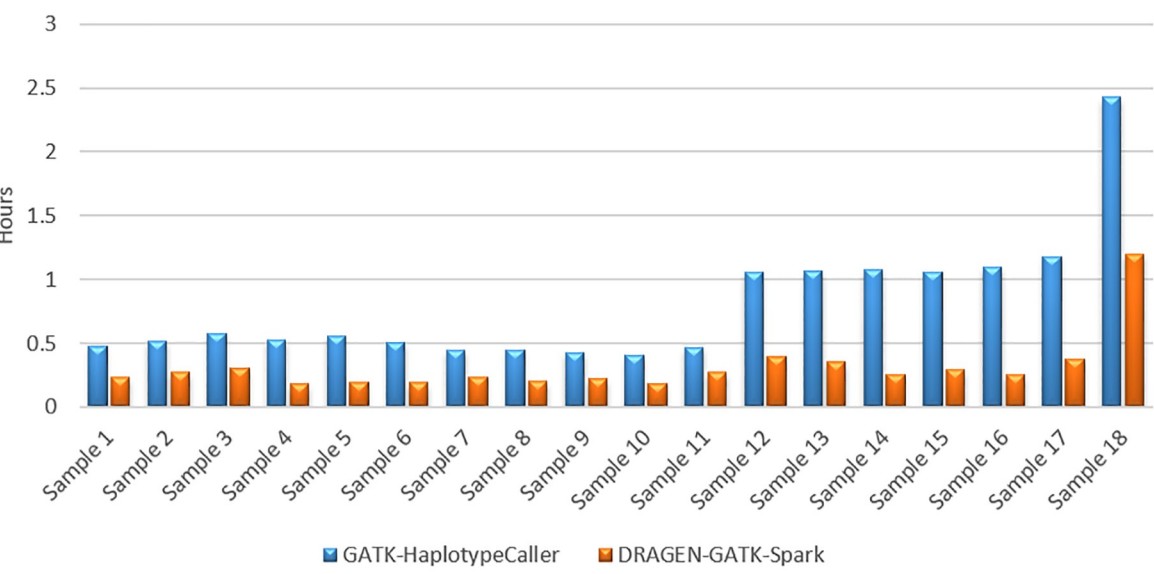

**Fig 6. Run time performance of the two variant callers.**

(GRCh37). The BWA-MEM2+DRAGEN-GATK pipeline processed germline mutation calling much more efficiently than the standard BWA+GATK pipeline, where the overall analysis time for different samples with different read counts and fastq file sizes decreased to half. Additionally, BWA-MEM2+DRAGEN-GATK is highly sensitive and precise in calling SNVs and indels. Overall, the focus of this study was to demonstrate the performance of the optimized version of the BWA-MEM2 aligner and the DRAGEN-GATK variant caller tools in detecting SNVs and indels compared to the modified versions of BWA-MEM2.

## Conclusion

We present a comprehensive evaluation between BWA-MEM 2+DRAGEN-GATK and GATK best practice workflows. All the code used for implementing the two pipelines is available on GitHub https://github.com/Abusamra/GATK–best-practice-vs-optimized-Pipeline-for-Improved-Detection-of-Germline-Exomes-. Based on the results and data analysis, we believe that using the BWA-MEM2 and the DRAGEN-GATK pipeline can be more accurate and efficient for the WES analysis.

## Acknowledgments

We would like to thank the Bioinformatics Unit of the Center of Excellence in Genomics Medicine Research and AZIZ Super-computing facilities at the High-Performance Computing Center at King Abdulaziz University for their help and technical support.

## Author Contributions

**Conceptualization:** Nofe Alganmi, Heba Abusamra.

**Data curation:** Nofe Alganmi, Heba Abusamra.

**Formal analysis:** Nofe Alganmi, Heba Abusamra.

**Funding acquisition:** Nofe Alganmi.

**Investigation:** Nofe Alganmi, Heba Abusamra.

**Methodology:** Nofe Alganmi, Heba Abusamra.

**Project administration:** Nofe Alganmi.

**Resources:** Nofe Alganmi, Heba Abusamra.

**Software:** Nofe Alganmi, Heba Abusamra.

**Supervision:** Nofe Alganmi.

**Validation:** Nofe Alganmi, Heba Abusamra.

**Visualization:** Nofe Alganmi, Heba Abusamra.

**Writing – original draft:** Nofe Alganmi, Heba Abusamra.

**Writing – review & editing:** Nofe Alganmi.

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
