## [Decision Letter · Decision Letter 0]

2 Jun 2022

PONE-D-22-13433Optimized Pipeline with Dragen and GATK Tools for Improved Detection of Germline ExomesPLOS ONE

Dear Dr. Alganmi

Thank you for submitting your manuscript to PLOS ONE. After careful consideration, we feel that it has merit but does not fully meet PLOS ONE’s publication criteria as it currently stands. Therefore, we invite you to submit a revised version of the manuscript that addresses the points raised during the review process.

We look forward to receiving your revised manuscript.

Kind regards,

Alvaro Galli

Academic Editor

PLOS ONE

Journal Requirements:

Reviewers' comments:

Reviewer's Responses to Questions

**Comments to the Author**

1. Is the manuscript technically sound, and do the data support the conclusions?

Reviewer #1: Partly

Reviewer #2: Partly

2. Has the statistical analysis been performed appropriately and rigorously? 

Reviewer #1: No

Reviewer #2: N/A

3. Have the authors made all data underlying the findings in their manuscript fully available?

Reviewer #1: No

Reviewer #2: No

4. Is the manuscript presented in an intelligible fashion and written in standard English?

Reviewer #1: Yes

Reviewer #2: No

5. Review Comments to the Author

Reviewer #1: This study evaluates a pipeline of variant calling for whole exome sequencing data using Dragen and GATK (named as Dragen-GATK), and BWA-MEM2. It compares the pipeline combined by BWA-MEM2 and Dragen-GATK with another combined by BWA-MEM and GATK-HaplotypeCaller in terms of variant calling performance and running time using four WES datasets. The authors state that the combination of BWA-MEM2 and Dragen-GATK performs better than the other combination.

There are some major concerns.

1. The contribution of this study in terms of pipeline development is quite minimal. Both BWA-MEM2 and Dragen-GATK were developed by other groups and they are already publicly available. As mentioned in this manuscript, it is already known that Dragen-GATK outperforms in variant calling. BWA-MEM2 is a newer version of BWA-MEM. It is not surprising that BWA-MEM2 and Dragen-GATK perform better that its previous version. What novel ideas were proposed to optimize BWA-MEM2 and Dragen-GATK in this study? There is no explanation what challenging issues the authors addressed in this study and with what ideas the issues were resolved.

2. The BWA-MEM2 and Dragen-GATK pipeline was evaluated using only four WES datasets. Is there any particular reason that the performance was evaluated using only four datasets analyzed by one consortium? If there is other WES datasets analyzed by other groups and they are regarded as a gold standard, they should be included in this study.

3. The pipeline available in the GitHub repository is not quite straightforward to install and run. More details about command lines to run and parameter setting information need to be provided to replicate the experiments performed in this study.

Reviewer #2: The text requires a detailed review of the language. Its current version contains a few grammatical errors that affects the comprehension of the text.

During submission, the authors answered a questionnaire stating that all data are fully available (without restriction). However, on the body of the text, the authors state that the reader must contact them (the authors) to get access to the data. This is not a good strategy for data sharing, as documented at the Data Availability recommendations page ( https://journals.plos.org/plosone/s/data-availability ). I would like to suggest the authors to use a public or institutional repository instead. Additionally, it is important that the aforementioned questionnaire has the correct information about the study: for example, the information about the Ethics Committee Authorization (IRB) isn’t properly described.

Alganmi and Abusamara describe a study where they use four well-described WES samples (NA24149, NA24143, NA24385, NA25631) and eighteen WES samples generated by their institution to compare the performance of two GATK-based pipelines. One pipeline combines the BWA-MEM aligner and GATK HaplotypeCaller; the other pipeline uses the BWA-MEM2 aligner and the Dragen-GATK (Spark enabled) caller. They assess performance by evaluating computing time and accuracy on the variant calls.

From the text, it seems that each set of FASTQ files (~0.5GB, 1GB, 2GB, 4GB) was analyzed only once by each pipeline. With this strategy, the readers do not have the information on the variability of processing times (e.g., is the time difference between .05GB and 2GB files significant?). Additionally, it is important to describe in details the origin of the time differences: are the reported diferences due exclusively to the use of Dragen-GATK? What portions of the gains are due to BWA-MEM vs. BWA-MEM2? Therefore, I would like to recommend the authors to compare the processing times at different phases of the pipelines (in particular, aligners and callers). Then, when creating the plots (one for the aligners, another for the callers and a third one for the total time), make sure that all data points (one for each run) are shown (instead of only the mean time).

The authors report improvements on indel calls. The Dragen-GATK pipeline called less than 100 indels per sample, while GATK called a few hundreds. Are the stochastic components in the Dragen-GATK pipeline that would allow different calls (not only indels) to happen for the same samples on different runs? If stochastic factors affect the calls, what are the average metrics (and standard-errors)? I understand the BWA-MEM2 can generate different results when compared to BWA-MEM, if mapping to ALT contigs; discussing this subject on the manuscript would improve it significantly.

The authors should also describe in details how/where their eighteen samples were used on the study. It seems they were used exclusively for the computing time assessment. Is this correct?

6. PLOS authors have the option to publish the peer review history of their article (what does this mean?). If published, this will include your full peer review and any attached files.

Reviewer #1: No

Reviewer #2: No

---

## [Author Response · Author response to Decision Letter 0]

11 Aug 2022

Dear Dr. Alvaro Galli, 

Thank you for giving me the opportunity to submit a revised draft of my manuscript titled “Optimized Pipeline with Dragen and GATK Tools for Improved Detection of Germline Exomes” to the PLOS ONE journal. We appreciate the time and effort that you and the reviewers have dedicated to providing your valuable feedback on our manuscript. We are grateful to the reviewers for their insightful comments on our paper. We have been able to incorporate changes to reflect on most of the suggestions provided by the reviewers. We have highlighted the changes within the revised manuscript.

Here is a point-by-point response to the reviewers’ comments and concerns.

Comments from Reviewer 1

Comment 1: The contribution of this study in terms of pipeline development is quite minimal. Both BWA-MEM2 and Dragen-GATK were developed by other groups and they are already publicly available. As mentioned in this manuscript, it is already known that Dragen-GATK outperforms in variant calling. BWA-MEM2 is a newer version of BWA-MEM. It is not surprising that BWA-MEM2 and Dragen-GATK perform better that its previous version. What novel ideas were proposed to optimize BWA-MEM2 and Dragen-GATK in this study? There is no explanation what challenging issues the authors addressed in this study and with what ideas the issues were resolved.

Response: Thank you for pointing this out. This work is novel because it is the first study according to our knowledge that evaluates the performance of the new optimized GATK tools and dragen and compares both the aligner and the variant caller as a complete pipeline to the standard best practice GATK pipeline. We incorporated the supercomputer environment in this work to utilize the optimized features BWA-MEM2 and Dragen-GATK. We are novel in testing these tools that we adapted using both well characterized genetics data and also real patient data to provide suggestions to any diagnostics lab that is performing WES tests. In the bioinformatics community, the benchmarking research is the only way to know if the tool/method is providing good results or not. The validations and performance assessment we have presented is very valuable to clinical bioinformatics research. 

In this regards, We suggest to adjust the paper title to be: “Evaluation of an Optimized Germline Exomes Pipeline using BWA-MEM2 and Dragen-GATK Tools”

Comment 2: The BWA-MEM2 and Dragen-GATK pipeline was evaluated using only four WES datasets. Is there any particular reason that the performance was evaluated using only four datasets analyzed by one consortium? If there is other WES datasets analyzed by other groups and they are regarded as a gold standard, they should be included in this study.

Response: Thank you for this suggestion. It would have been interesting to explore many standard datasets. However, in the case of our study, it seems slightly not necessary because the GIAB is a scientific consortium that is publicly known and trusted. There are standard controls available from private companies. But the GIAB samples are well characterized controls that are widely used and recommended by many genetics organizations for diagnostic validation as it covers all the areas in the genome that we like to validate. Therefore, we don't need any more samples to trust the results. Another issue is the cost, we bought the GIAB samples and ran them as WES experiments in our NGS lab using a NovaSeq machine (we pay for the kits and reagents) which is expensive. It might not be feasible for many labs to run large amounts of samples if they provide the same information. 

From GIAB, we tested four samples: three of them represent a family and the fourth is from a different population. We also did reproducibility study for these samples and ran them many times to make sure we are getting robust results. Finally we show the time performance across real NGS data to reflect the variability of the FASTQ files generated using gemline real samples to evaluate the bioinformatics tools.

Comment 3: The pipeline available in the GitHub repository is not quite straightforward to install and run. More details about command lines to run and parameter setting information need to be provided to replicate the experiments performed in this study.

Response: We agree with this and have incorporated your suggestion by creating a readme in our GitHub site as follows:

https://github.com/Abusamra/GATK--best-practice-vs-optimized-Pipeline-for-Improved-Detection-of-Germline-Exomes-

However, As this work runs on a supercomputer environment, a good linux background is needed.

Comments from Reviewer 2

Comment 1: [The text requires a detailed review of the language. Its current version contains a few grammatical errors that affects the comprehension of the text.] 

Response: Agree. We have, accordingly, revised all grammatical errors and all have been corrected. It was revised by prof. Mohammed Alqahtani (https://mhalqahtani.kau.edu.sa/CVEn.aspx?Site_ID=0004963&Lng=EN) who is a full professor in genetics at king abdulaziz university. He was very keen to track all the grammatical mistakes in the manuscript.

Comment 2: [During submission, the authors answered a questionnaire stating that all data are fully available (without restriction). However, on the body of the text, the authors state that the reader must contact them (the authors) to get access to the data. This is not a good strategy for data sharing, as documented at the Data Availability recommendations page ( https://journals.plos.org/plosone/s/data-availability ). I would like to suggest the authors to use a public or institutional repository instead.]

Response: Agree. We are sorry that the conflict happened. We intended to not put the dataset in a public repository due to more research we are still doing. However, after getting your comments we are happy to share all the datasets. So now all datasets will be publicly available online. 

Comment 3: [Additionally, it is important that the aforementioned questionnaire has the correct information about the study: for example, the information about the Ethics Committee Authorization (IRB) isn’t properly described.] 

Response: Apologies for this. We would like to share every information we have. The ethical committee that approved this research is called “CEGMR Bioethics Committee '', for more information about the committee here is the official website: https://cegmr.kau.edu.sa/Content-117-EN-238247. Also here is the head of the committee website: https://sa.linkedin.com/in/adeel-chaudhary-64824615. I will make sure if it is possible to update the questionnaire information during the submission of the revised manuscript. 

If there is a particular information needed kindly help us to know it and we will update.

Comment 4: From the text, it seems that each set of FASTQ files (~0.5GB, 1GB, 2GB, 4GB) was analyzed only once by each pipeline. With this strategy, the readers do not have the information on the variability of processing times (e.g., is the time difference between .05GB and 2GB files significant?).

Response:Thank you for pointing this out. We like to clarify that what has been reported in the paper was actually the average of running time for each sample about 20 times. The aim was to make sure the time value was representing each of the file sizes. For each sample we found a difference of a few minutes thus we took the average time for each sample with each pipeline. We pay your attention that we fixed all of the pipeline settings and hardware for each time we run the 18 samples. We mean that the 18 samples started running at the same time and this was repeated 20 times. We wanted to eliminate the differences that could arise due to some technicality issues with the supercomputer system. We made sure that all have the same settings. This was highlighted in the revised manuscript at page 7, line 163.

Comment 5: Additionally, it is important to describe in details the origin of the time differences: are the reported diferences due exclusively to the use of Dragen-GATK? What portions of the gains are due to BWA-MEM vs. BWA-MEM2? Therefore, I would like to recommend the authors to compare the processing times at different phases of the pipelines (in particular, aligners and callers). Then, when creating the plots (one for the aligners, another for the callers and a third one for the total time), make sure that all data points (one for each run) are shown (instead of only the mean time).

Response: We agree with this and have incorporated your suggestion throughout the revised manuscript. This was highlighted in the revised manuscript pages 3 and 6.

Comment 6: The authors report improvements on indel calls. The Dragen-GATK pipeline called less than 100 indels per sample, while GATK called a few hundreds. Are the stochastic components in the Dragen-GATK pipeline that would allow different calls (not only indels) to happen for the same samples on different runs? 

If stochastic factors affect the calls, what are the average metrics (and standard-errors)? 

We agree with this and have incorporated your suggestion throughout the manuscript. We assessed the reproducibility of the variants for both SNVs and indels called by the BWA-MEM 2+Dragen-GATK pipeline using the GIaB samples (NA24149, NA24143, NA24385,and NA24631) in four replicated runs in two perspectives:

1) The performance results (TP,FP,FN,recall, and precision) of the four replicated results for each sample were identical (Table 3 in the revised manuscript). 

2) the common and differences of variant detection among the four replicates. Venn diagram were used to illustrate the number of variants detected for different replicate for each sample (figure3 in the revised manuscript), and concordance of the variants among replicated runs were measured (99.9%) as well as the error rate= [(|Approximate Value - Exact Value|) / Exact Value] x 100 was calculated with the result of (0.019%),(0.002%),(0.09%), (0.07%) for the NA24149, NA24143, NA24385,and NA24631 samples respectively.

Comment 7: I understand the BWA-MEM2 can generate different results when compared to BWA-MEM, if mapping to ALT contigs; discussing this subject on the manuscript would improve it significantly.

Response: Thank you for this suggestion. It would have been interesting to explore this aspect. However, in the case of our study, it seems slightly out of scope because the ambiguous mapping is highly raised with human genome reference hg38 which we are not using in our lab or in this study. We are using hg19 . Another reason, this ALT contig represents the variability which we are trying to avoid in our research for the clinical diagnostic. We are focusing on showing the performance of the bioinformatics tools on consistent pipelines. We plan to add this in our future work as it is beneficial to genetics variant detection if the population carries alternate haplotypes but this will need different setting and study design,

Comment 8: [The authors should also describe in details how/where their eighteen samples were used on the study. It seems they were used exclusively for the computing time assessment. Is this correct?] 

Response: Correct. This was mentioned in the abstract as follows: “In addition, eighteen exome samples were analyzed based on different read counts and coverage were used precisely for the run time assessment.“

Also in the methodology section we explain why we included the real patient data and for what they were used:

“In addition we collected real data from eighteen patients who were sequenced at Genati labs (which is a College of American Pathologists accredited laboratory) in the Center of Excellence in Genomic Medicine Research (CEGMR) at King Abdulaziz University. This study was approved by IRB no. 32-CEGMR-Bioeth-2021. Since the four reference samples from GiaB were of the same size in yields and read counts perspective, we selected the eighteen exome sequencing files to be of different yields, coverage, and mapped reads to evaluate the pipelines’ time performance.”

Then we explained the analysis we performed:

“To facilitate the time performance comparison, these samples were distributed into four groups based on fastq data yield. Six samples were of the smallest amount yield (2-3 GB), eight samples of 4-6 GB yield, 3 samples of 8-11 GB yield, and one sample with largest yield of 17 GB. Table 1, provides detailed information of these eighteen data sets grouped based on fastq data yield, passing filter (PF) clusters for both lane 1 and lane 2.“

Additional clarifications

In addition to the above comments, all grammatical errors pointed out by the reviewers have been corrected. We also suggest to adjust the title based on the reviewer comments to be “Evaluation of an Optimized Germline Exomes Pipeline using BWA-MEM2 and Dragen-GATK Tools”

We look forward to hearing from you in due time regarding our submission and to respond to any further questions and comments you may have.

Sincerely, 

Nofe Alganmi

17/7/2022

---

## [Decision Letter · Decision Letter 1]

27 Sep 2022

PONE-D-22-13433R1Evaluation of an optimized germline exomes pipeline using BWA-MEM2 and Dragen-GATK toolsPLOS ONE

Dear Dr. Alganmi,

Thank you for submitting your manuscript to PLOS ONE. After careful consideration, we feel that it has merit but does not fully meet PLOS ONE’s publication criteria as it currently stands. Therefore, we invite you to submit a revised version of the manuscript that addresses the points raised during the review process.

We look forward to receiving your revised manuscript.

Kind regards,

Alvaro Galli

Academic Editor

PLOS ONE

Reviewers' comments:

Reviewer's Responses to Questions

**Comments to the Author**

1. If the authors have adequately addressed your comments raised in a previous round of review and you feel that this manuscript is now acceptable for publication, you may indicate that here to bypass the “Comments to the Author” section, enter your conflict of interest statement in the “Confidential to Editor” section, and submit your "Accept" recommendation.

Reviewer #1: (No Response)

Reviewer #3: All comments have been addressed

2. Is the manuscript technically sound, and do the data support the conclusions?

Reviewer #1: No

Reviewer #3: Yes

3. Has the statistical analysis been performed appropriately and rigorously? 

Reviewer #1: N/A

Reviewer #3: Yes

4. Have the authors made all data underlying the findings in their manuscript fully available?

Reviewer #1: Yes

Reviewer #3: Yes

5. Is the manuscript presented in an intelligible fashion and written in standard English?

Reviewer #1: Yes

Reviewer #3: Yes

6. Review Comments to the Author

Reviewer #1: There are some improvements in this revision. There is still a major concern that has not been resolved yet in terms of the contribution of this work.

There is a web page about a whole genome germline pipeline for variant discovery, which is provided by the Dragen-GATK team (https://app.terra.bio/#workspaces/warp-pipelines/DRAGEN-GATK-Whole-Genome-Germline-Pipeline). According to the page, there is their own short-read mapping tool (called DRAGMAP aligner). It was released in 2021. Is there any particular reason that the DRAGMAP aligner was not included in this evaluation study?

A manuscript which was uploaded recently in biorxiv (https://www.biorxiv.org/content/10.1101/2022.09.18.508404v1.full) concludes that the Dragen-GATK pipeline including the DRAGMAP aligner performs better than 'BWA-MEM2 + Dragen-GATK variant caller'.

Reviewer #3: In this paper, the authors are proposed “Evaluation of an optimized germline exomes pipeline using BWA-MEM2 and Dragen- GATK tools”

The strengths of the paper are that it is well structured, the description of the related work is well done and that results are extensively compared to results of the similar research.

Minor revisions:

1. Authors draw a graphical abstract of this study

2. Proofread the entire manuscript

3. Authors should confirm the title of the manuscript, because in manuscript one title is there in response to reviewer one title name is there.

4. Authors should include the discussion part, In that they have to discuss the novelty of the proposed approach.

7. PLOS authors have the option to publish the peer review history of their article (what does this mean?). If published, this will include your full peer review and any attached files.

Reviewer #1: No

Reviewer #3: No

---

## [Author Response · Author response to Decision Letter 1]

9 May 2023

Dear Reviwer 1,

Thanks for your valuable comments. Allow us to respond to your following questions:

1) you asked about a similar work that compare two aligners, however their work was in whole genome settings while our work handled the whole exome data which more useful for clinical genetics testing routine. Secondly, you asked if we excluded the DRAGMAP aligner and the answer is no because DRAGMAP aligner was in ALPHA version when we start this work and we tried it but it always gave errors, Thus we decided to give more time before we publish its results (it might not reliable). Here is the query we issued to illumina team in this regards you can see the name Abusamra ,one of the authors, has asked the question:

https://github.com/Illumina/DRAGMAP/issues/7

Also the paper you mentioned they didn't use the DRAGMAP aligner.!

I hope that this paper you mentioned proof how valuable is the work we are doing in terms of bioinformatics tools that will be used in clinical setting. In terms of exome which is more meaningful in clinical uses compared to whole genome which is more used for research applications.

---

## [Decision Letter · Decision Letter 2]

27 Jun 2023

Evaluation of an optimized germline exomes pipeline using BWA-MEM2 and Dragen-GATK tools

PONE-D-22-13433R2

Dear Dr. Alganmi,

We’re pleased to inform you that your manuscript has been judged scientifically suitable for publication and will be formally accepted for publication once it meets all outstanding technical requirements.

Kind regards,

Alvaro Galli

Academic Editor

PLOS ONE

Additional Editor Comments (optional):

Reviewers' comments:

Reviewer's Responses to Questions

**Comments to the Author**

1. If the authors have adequately addressed your comments raised in a previous round of review and you feel that this manuscript is now acceptable for publication, you may indicate that here to bypass the “Comments to the Author” section, enter your conflict of interest statement in the “Confidential to Editor” section, and submit your "Accept" recommendation.

Reviewer #1: All comments have been addressed

2. Is the manuscript technically sound, and do the data support the conclusions?

Reviewer #1: Yes

3. Has the statistical analysis been performed appropriately and rigorously? 

Reviewer #1: Yes

4. Have the authors made all data underlying the findings in their manuscript fully available?

Reviewer #1: Yes

5. Is the manuscript presented in an intelligible fashion and written in standard English?

Reviewer #1: Yes

6. Review Comments to the Author

Reviewer #1: The authors have addressed all my concerns and comments adequately. I recommend publication in PLOS ONE.

7. PLOS authors have the option to publish the peer review history of their article (what does this mean?). If published, this will include your full peer review and any attached files.

Reviewer #1: No

---

## [Editor Report · Acceptance letter]

26 Jul 2023

PONE-D-22-13433R2 

Evaluation of an optimized germline exomes pipeline using BWA-MEM2 and Dragen-GATK tools 

Dear Dr. Alganmi:

I'm pleased to inform you that your manuscript has been deemed suitable for publication in PLOS ONE. Congratulations! Your manuscript is now with our production department. 

Kind regards, 

on behalf of

Dr. Alvaro Galli 

Academic Editor

PLOS ONE